# *Fusarium* spp. in Metalworking Fluid Systems: Companions Forever

**DOI:** 10.3390/pathogens13110990

**Published:** 2024-11-13

**Authors:** Célia Ruiz, Giulia von Känel, Stefan Burkard, Peter Küenzi

**Affiliations:** Department of Microbiology, Blaser Swisslube AG, 3415 Hasle-Rüegsau, Switzerland; celiaruizo6@gmail.com (C.R.);

**Keywords:** fungi, *Fusarium* spp., metalworking fluids, flow cytometry, occupational health

## Abstract

Water-miscible metalworking fluids (MWFs) are utilized in a variety of metal removal and forming operations. For end-use, formulation concentrates are diluted in water, creating conditions conducive to microbial growth and metabolism, possibly compromising the fluid’s integrity and mechanically obstructing filters or piping systems. Metalworking machines offer additional habitats on surfaces that are in permanent or temporary contact with MWFs. For that reason, biocides have been incorporated into concentrates for years, but legal constraints will restrain their use in the future. While bacterial contamination of MWFs is well documented, fungal contamination is often overseen and infrequently reported in the literature. In this study, we report fungal prevalence in in-use MWFs sampled worldwide over 10 years, and we are convinced that the presence of fungi is the norm rather than the exception. In addition, we evaluated the inhibitory effect of fungicides on fungal growth, sporulation and spore viability using traditional culture-dependent methods and flow cytometry. In essence, we show that the effectiveness of these fungicides is limited and dependent on the chemical construction of the fluid. We think that the ecology created by water-diluted MWFs is of higher importance than the anti-fungal activity of single components.

## 1. Introduction

Water-miscible metalworking fluids (MWFs) are used to cool and lubricate during metal removal and forming operations in a plethora of applications. MWFs are formulated and sold as concentrates that contain anything from 10 to 20 organic ingredients, mixed at the end user’s site with water that subsequently accounts for 85% to 95% of the mixture [1]. There are three main classes of MWFs: emulsifiable oils, semisynthetics and synthetics. The main components of MWF concentrates are mineral and ester oil (sourced from plants), polyalphaolifins or glycols. To improve performance, stability and functionality, emulsifiers, corrosion inhibitors, foam control agents and lubricity enhancers are added as needed. In addition, biocidal and biostatic components are critical ingredients of MWF formulations [2,3] as they help to keep microbiological agents under control. Still, the high ratio of water and the evenly mixed-in organic components, aerated by recirculation, provide a decent base of life for many planktonic bacteria, fungi and archaea in all types of MWFs [4]. Additional habitats are provided on (machine) surfaces in temporary contact with MWFs by means of splashing, evaporation and misting. Lines that supply and discharge the MWF from the site of action are often only partially filled and offer dozens of square meters of microbial settlement area. This is a considerable problem with single-filled machines, which become multiplied in centralized systems where the fluid is transported over long distances to and from many machining centers [5]. Microbial growth combined with metal chips and swarfs leads to clogging of filters and residue formation. Therefore, biofilms and fungal growth on surfaces are probably of far greater importance than their planktonic relatives and much more difficult to maintain [6].

Fungi are eukaryotic, aerobic organisms, which are regularly detected in MWFs and their systems [7]. Morphologically, fungi can be subdivided into multicellular, filamentous molds or unicellular yeast. However, dimorphism, the ability to switch from filamentous growth to a unicellular lifestyle and vice versa, has been described for some fungi, making visual identification difficult [8,9,10]. In the past, the presence of yeasts in MWFs was thought to be unusual and generally associated with physico-chemical instabilities, but later findings showed that some yeasts (*Candida* spp., *Yarrowia* spp.) do survive and multiply in MWFs even at a pH of 9 or higher [11] and can be successful inhabitants.

Molds grow as multicellular filaments called hyphae, which form mycelia [12]. Molds are important types of fungi, represented by a large number of species that play key roles in the breakdown of organic matter [13]. Molds propagate themselves through the production of spores at the head of special aerial hyphae that are easily transported by air and fluids [14]. In MWF systems, molds are mainly found on surfaces such as splash zones or in areas that are moistened by evaporation and misting. Consequently, their presence becomes evident when spores have been discharged into the fluidic phase or if fragments have become detached. Costly downtimes and serious technical consequences can result if molds manage to colonize submerged surfaces or if parts of them are washed off and carried away as filters clog, and MWF circulation stops [4]. Molds reportedly recovered from MWFs include members of the genera *Aspergillus*, *Fusarium*, *Exophiala*, *Trichoderma*, *Cladosporium* and *Penicillium* [7,11,15,16]. In our experience, molds isolated from in-use MWFs predominantly belong to the genus *Fusarium*.

While it used to be common for the detection of molds to be associated with low pH values, this is no longer the case today; this situation is comparable to that of yeasts. The main source is quite likely spores transported through the air, as the genus *Fusarium* is widely distributed in soil and often associated with agriculturally important plants, causing crop diseases [17,18]. Some species, like *Fusarium solani* species complex (*FSSC*), have been reported to induce a range of diseases and infections in immunocompromised and healthy human beings as well in animals, mostly of aquatic species. For these reasons, FSSC has been included within pathogens related to the One Health issue [19,20]. FSSC (Phylum Ascomycota, Class Sordariomycetes, Order Hypocreales; Family Nectriaceae) is a group estimated to contain at least 60 phylogenetically distinct species [21].

To prevent fungal contamination and growth, most MWFs are protected by the incorporation of dedicated pesticides, commonly known as fungicides. However, regulatory pressure on these chemicals is increasing, leading to significant restrictions on permitted concentrations in both the concentrate and in-use products, a trend that is expected to continue [22]. The range of potential components has already narrowed down to four: Sodium Pyrithione (NaPT), Butyl-benzisothiazolinone (BBIT), Octylisothiazolinone (OIT) and Ortho-phenyl Phenol (OPP), of which only two (NaPT and BBIT) are practical solutions for in-drum conservation. OIT has a short shelf-life in concentrates and is therefore generally used as a tank-side additive, while the solubility of OPP makes successful incorporation into many formulations challenging.

In this study, we report on fungal prevalence in in-use MWFs sampled worldwide over 10 years from 2014 to 2023. According to our findings, fungal contamination of MWF systems is the norm rather than the exception, especially regarding filamentous fungi surviving on machine surfaces. In contrast, directly measurable occurrence in MWFs is rare. That technical issues do not go out of hand in the industry is often attributed to fungicides incorporated into the base formulation or added at the tank side.

In laboratory experiments, we thus evaluated the inhibitory effect of fungicides and their vapors on fungal growth, sporulation and spore viability using traditional culture-dependent methods and flow cytometry. In essence, we show that the effectiveness of these fungicides is limited and dependent on the chemical composition of the MWF. Sooner rather than later, this industrial sector needs to learn how to create and use MWFs without these dedicated compounds.

## 2. Materials and Methods

### 2.1. Access to MWF and Residue Samples

Via our extensive customer service network, we had worldwide access to MWF and residue samples from end users, both our own and external customers.

### 2.2. Fungicides

For all experiments, technical standard fungicides from industrial suppliers were used; the concentration indicated refers to a typical dose in a freshly prepared, 5% (*w*/*w*) in-use MWF, i.e., NaPT (Acticide^®^ LV 508; Thor GmbH, Speyer, Germany; Active ingredient content: 40%) at a concentration of 250 ppm; BBIT (Densil^®^ DN; Arch Biocides, Atlanta GA, USA; Active ingredient content: 100%) at a concentration of 50 ppm; and OPP (PREVENTOL^®^ O extra; Lanxess Deutschland GmbH, Leverkusen, Germany; active ingredient content: 99.5%) at a concentration of 500 ppm.

### 2.3. Metalworking Fluids

Most of the experiments shown were carried out with two experimental MWFs based on mineral oil that share about 50% of the ingredients (MWF A and B). Both concentrates were prepared available either with or without both NaPT and BBIT; the fungicide concentration was as described above when diluted at 5% (*w*/*w*).

### 2.4. Examination and Isolation of Fungi from MWF Samples

Isolation of fungal species was based on cultivation-dependent methods such as the standard heterotrophic plate count method (HPC) or the use of dip slides. HPC was performed on Sabouraud dextrose–agar (SDA) prepared in-house (Oxoid #CM041R; Thermo Fisher, Prattlen, Switzerland) at 65 g L^−1^. A total of 50 μL of either undiluted or 1:100 diluted 0.9% NaCl was plated using an Eddy Jet 2 system (IUL, Barcelona, Spain) in logarithmic mode. The plates were subsequently incubated at 30 °C for a minimum of 72 h before analysis. Fungal species were identified by Maldi-TOF MS analysis of protein patterns at Mabritec AG (Riehen, Switzerland). As dip slides, Cult Dip combi^®^ (Millipore #1.00778.0001; Merck, Darmstadt, Germany), which offers a Rose Bengal Agar, were used according to the manufacturer’s instructions. The strain of *F. solani* used in the following experiments stemmed from these samples.

### 2.5. Examination of Residue Samples

Residue samples were either examined directly by conventional light microscopy, or, if opaque and/or semi-solid, stained with Calcofluor-White (Merck #18909, Darmstadt, Germany) and examined by fluorescence microscopy (Olympus BX43 equipped with a reflected fluorescence system; Olympus Europa, Hamburg, Germany). Calcofluor-White is a fluorescent blue dye which binds to 1,3 and 1,4-beta-polysaccharides [23].

### 2.6. Zone-of-Inhibition Tests

Zone-of-inhibition tests were performed as described [24]. Briefly, isolated fungal samples were dissolved in 1 mL of 0.9% NaCl and 200 µL of the resulting mixture evenly spread on an SDA. A small Whatman filter paper (ø 2 mm) was placed into the center of the plate and 1 µL of the undiluted fungicide subsequently added to the filter disk. Plates were incubated at 30 °C for 4 days.

### 2.7. Adaptation to Fungicides

SDA plates were prepared as described above, but shortly before solidifying, different concentrations of NaPT (0, 250 ppm, 500 ppm) or BBIT (0, 50 ppm, 100 ppm) were added and evenly mixed in. Plates were left to cool completely before being cut into thirds and reassembled. At the beginning of the experiment, spores and mycelium parts of *F. solani* were suspended in 0.9% NaCl and spread on the starting third containing no fungicides and cultivated overnight at RT. Plates were evaluated after 1, 2 and 3 weeks.

### 2.8. Sporulation Assays

Sporulation assays were performed based on the publication by Zhang et al. [25]. As background media, 0.9% NaCl was used and buffered with 50 mM TAPS (Sigma #T5130; Merck, Darmstadt, Germany), and the pH was adjusted to 9.3 with NaOH.

To start the assay, a round piece of SDA (ø 1.4 cm; 1.56 cm^2^) overgrown with *F. solani* after an incubation period of 4 days was added to 50 mL of buffer in a 250 mL baffled Erlenmeyer glass flask (SCHOTT DURAN, Mainz, Germany) and incubated on a shaker at 80 rpm at RT for 66 to 72 h. The resulting spore-containing solution was subsequently decanted, vortexed and distributed into 15 mL centrifuge tubes (Corning; #430791; Reynosa, Mexico). Fungicides or MWFs, alone or in combination at the indicated concentrations were subsequently added. The end volume was 5 mL.

### 2.9. Stability Assays

For long-term experiments, MWFs were premixed in TAPS-buffered saline at 5% (*w*/*w*), and 50 mL was added to 250 mL baffled Erlenmeyer flasks before adding the 1.56 cm^2^ piece of fungus-overgrown SDA. To give the fungus a chance to survive and develop, nutrients in the form of tryptone soy broth (Oxoid #CM131; Thermo Fisher, Pratteln, Switzerland) were added in 0.1-fold concentration right at the start.

### 2.10. Quantification of Spores and Viability Assays

For assays without MWFs, 200 µL aliquots were removed and directly stained with propidium iodide (PI) and SYTO9 (LIVE/DEAD™ BacLight™ Bacterial Viability and Counting Kit; Invitrogen L34856; Thermo Fisher, Pratteln, Switzerland) as described by Vanhauteghem et al. [26]. Analysis was subsequently performed on a CytoFLEX S flow cytometer (Beckman Coulter International S.A., Nyon, Switzerland).

Samples containing MWFs had to be cleaned by centrifugation prior to analysis; 200 µL was added to 1 mL of 5.25% Nycodenz^®^ (Serumwerk Bernburg #18003; Bernburg, Germany) with TAPS-buffered saline into 2 mL centrifuge tubes (Eppendorf #0030 123.344; Hamburg, Germany), mixed by vortexing and centrifuged at 10,000× *g* for 10 min. at 4 °C. The supernatant was removed by decantation and left to drain for a few minutes. Leftover MWF sticking to the sidewalls of the centrifugation tubes was removed with sterile cotton swaps before the pellet was re-dissolved in 200 µL TAPS-buffered saline, stained and analyzed as described above.

### 2.11. Volatilization of Fungicides-Assays

SDA plates were inoculated with four 20 µL drops of *F. solani* dissolved in 0.9% NaCl and incubated overnight at RT. The next day, 300 mL of MWF, mixed in with sterile-filtered tap water at 5% (*w*/*w*), was supplied to 600 mL sterile glass beakers (VWR International #213-1126, Dietikon, Switzerland) containing a sterile stirring bar. Then, the pre-incubated SD was added upside-down to the beaker and secured to be air-tight using Parafilm “M” (Amcor PM996, Zürich, Switzerland). The beakers were subsequently incubated with constant stirring on a heated, magnetic stirrer (30 °C, 200 rpm) for one week.

## 3. Results

### 3.1. Fungal Occurrence in MWFs and Machining Systems

#### 3.1.1. In MWFs

In the period from 2014 to 2023, we analyzed a total of 48,695 liquid MWF samples for the presence of fungi using cultivation-dependent methods. Samples included mineral oil- or vegetable oil-based MWFs, synthetic and semisynthetic products from end users around the world. In 5.6%, or a total of 2746 samples, we detected fungi either in the form of spores, hyphae or yeast cells, ranging from a few cells to thousands per milliliter. The frequency of detection did not change significantly over these 10 years (Table 1), indicating that the presence of fungi remained constant despite a substantial development in MWF technology [27]. What had changed, however, were the circumstances under which fungi were detected; ten years ago, detection was positive if the pH value was well below the recommended value for the MWF in question. In recent years, detection was also positive even if the pH value was still within the recommended range. 

#### 3.1.2. In Residue Samples

We only received a total of 417 residue samples for analysis from 2014 to 2023, as it seemed unreasonable for customers to send them in. On average, more than 50% of all samples analyzed contained yeast cells, hyphae, and/or spores (Table 1). However, we are aware that many deposits were not recognized as fungi or that they were hidden: Most metalworking machines are not designed to allow sufficient examination of their surfaces. Often, only the tank and the inside of the machining area are freely accessible, but most surfaces are only accessible after dismantling the machine enclosure. For this reason, fungal deposits were only detected and sent in if there was already a strong suspicion of colonization and even then, many residues were simply photographed and subsequently disposed of. However, photographs do not allow an accurate identification [4]. Still, as the reasons for sampling from customers or field staff have hardly changed over these years, we assume that the frequency of fungal contamination remains constant.

#### 3.1.3. Detected Species

As fungi are generally undesirable in MWFs, we did not carry out species identification regularly. However, before performing resistance tests (see Section 3.2.1), we determined the species upstream using Maldi-TOF MS (Mabritec AG, Riehen, Switzerland). The overwhelming majority of mold samples identified in this way belonged to the genus *Fusarium* (Table 2), which could mainly be assigned to either the *F. solani* or the *F. oxysporum* species complex. As these species are ubiquitous in nature, it is reasonable to assume that spores from the respective environment were transported by air into machine interiors and settled under suitable conditions, as shown for many indoor fungi [28]. We can only speculate on the origin of the isolated yeast samples. *Diutina neorugosa* and *D. rugosa* (formerly *Candida neorugosa* and *C. rugosa*) belong to a complex of species that represent about 0.2% of all clinical isolates [29], whereas *C. tropicalis* is one of the most frequently involved non-albicans *Candida* pathogens [30]. Apart from that, little is known, but it has been reported that these species are also ubiquitous in the environment [31,32]. *Yarrowia lipolytica* is also thought to be widespread in nature but is best-known as a frequently used, strictly aerobic, non-pathogenic producer of biodiesel [33] and other biotechnological products.

### 3.2. Escape from Fungicide-Toxicity in MWFs

Most MWFs contain fungicides to prevent contamination by fungi. Interestingly, the majority of isolated or detected species in this study originated from MWFs formulated with one or even two fungicides, most often BBIT and/or NaPT. With zone-of-inhibition tests, we investigated the efficacy of these fungicides on fungal growth.

#### 3.2.1. Resistance Formation in In-Use Samples

A total of 103 samples were tested for susceptibility to BBIT or NaPT, whether they had been previously exposed to these substances by means of the MWF (as ingredient(s) of its base formulation) or not. Fungi originating from MWFs preserved by NaPT showed a reduced susceptibility when re-exposed to NaPT. This is reflected in a significantly reduced zone of inhibition. Interestingly, such an effect was not observed with BBIT (Figure 1a).

#### 3.2.2. Adaptation to NaPT and BBIT

In the next step, we tested if a naïve laboratory strain of *F. solani* can adapt to NaPT or BBIT. To do this, we prepared SDA plates with different amounts of fungicide, which we then assembled in a way that the fungus was not exposed to any fungicide in the first third before it was subsequently exposed to a single and finally a double dose (Figure 1b). To give the fungus a head-start, it was spread on the starting third beforehand and cultivated overnight at RT. Only then were the agar thirds with the fungicides added. While *F. solani* in the control (without fungicides) and the BBIT-experiments had moved across the entire plate within 3 weeks with no discernible differences, only very modest growth was observed in the presence of NaPT. NaPT was apparently potent and even diffused into the first third initially containing no active ingredient.

### 3.3. Impact of Fungicides on Sporulation

#### 3.3.1. Quantities of Released Spores

As fungal growth and sporulation are distinct processes, we aimed to investigate the impact of fungicides and MWFs specifically on sporulation. In a recent review, the authors reported that the sporulation of *Fusarium* spp. can be inhibited or induced by a variety of chemical molecules [34], while others described that highest sporulation quantities in liquids are to be expected in saline at a pH between 9 and 10 [25]. Thus, in a simple experiment, fungicides were dissolved in TAPS-buffered saline at a pH of 9.3; the concentration used basically corresponded to that being present in many commercial in-use MWFs at 5% (*w*/*w*). To start the experiment, a 1.56 cm^2^ piece of SDA completely overgrown by *F. solani* was added to it. Between 5.7 and 6.6 log_10_ of spores were released by this small piece. As shown in Table 3, none of the fungicides had any significant impact on sporulation as counted by flow cytometry, either alone or in combination.

We also tested the influence of MWFs on sporulation and conducted experiments with MWF A and B in the absence and presence of added fungicides; again, sporulation remained unchanged.

#### 3.3.2. Effect of Fungicides on Spore Viability

To assess the viability of the released spores, we removed aliquots from the control experiment before adding fungicides NaPT, BBIT or a combination of both. We then assessed spore viability by flow cytometry using Syto9 and PI as described by Vanhauteghem et al. [26] after incubation for 30 min, 4 h and 24 h. Whereas BBIT reduced spore viability gradually and efficiently (Figure 2a), NaPT had almost no impact and was largely indistinguishable from the control. A combination of both fungicides was marginally less effective than BBIT alone. This suggests that NaPT affects the fungus itself (see Section 3.2.2) but not the spores once released. A simple test supported this assumption; NaPT or BBIT were added to the buffer system before adding the overgrown piece of Sabouraud agar, and they were incubated at RT with shaking (80 rpm). After 24 h, the viability of the spores was assessed as described, showing a clear but incomplete reduction in the presence of NaPT (Figure 2b).

#### 3.3.3. Effect of MWFs on Spore Viability

To test the efficacy of NaPT and BBIT in conjunction with MWFs, we repeated viability tests with MWF A and B. These experiments showed that MWF A deactivated virtually all spores after 24 h, while MWF B did not (Figure 3). As expected, this measurable effect was enhanced by fungicides in both cases; MWF A thus deactivated more than 40% of the spores after only 4 h, while the value for MWF B was around 25%.

#### 3.3.4. Effect of Delayed Spore Inactivation

Although the result after 24 h was identical in terms of spore inactivation for MWF A and B (both with added fungicides), the delay in the process could lead to problems in practice, particularly when these spores are splashed onto surfaces where they only occasionally remain in contact with the MWF. To simulate such a situation, we prepared 50 mL emulsions of MWF A and B, respectively, including fungicides at 5% (*w*/*w*) in 250 mL baffled Erlenmeyer flasks and added a 1.56 cm^2^ piece of SDA, overgrown with *F. solani.* After one week (80 rpm, RT), MWF B began to exhibit instabilities, as evidenced by a creamy layer forming on the surface (Figure 4a). By three weeks, a visible deposited layer appeared along the contact line (Figure 4b). In contrast, MWF A showed only emulsion components, few spores, and detached hyphae. MWF B, however, allowed isolated spores and hyphae to survive, leading to the formation of detectable mycelium at and above the contact line (Figure 4c). In real-world systems, turbulence is unavoidable and transports spores and hyphae to surfaces where they are only partially washed away. This results in temporary and incomplete toxic effects, thereby enabling fungal survival. For this experiment, the MWFs were diluted in TAPS-buffered saline, and pH changes remained minimal.

### 3.4. Impact of Fungicide Volatiles

As mentioned, fungi are mainly found on machine surfaces outside the MWF and only temporarily in direct contact with the fluid. Typical locations are, for example, in the pump area where the fluid level remains practically constant. Such sections are usually covered to minimize evaporation into the production hall, which leads to high local humidity in the section itself, promoting fungal growth [4]. Accordingly, such areas cannot be decontaminated by simply adding fungicides, unless these are effective in the vapor phase. We therefore developed a laboratory-scale test to examine this possibility (Figure 5a). After one week, there were no significant effects on fungal growth on the SDA plate, regardless of the chemical composition or the presence of NaPT and/or BBIT. However, we conducted additional experiments using OPP at a concentration of 500 ppm, which consistently showed a significant and detrimental impact on the development of *F. solani* on the SDA plate above, independent of the chemical background of the MWFs used. Figure 5b illustrates experiments with MWF A both without fungicides and with NaPT, BBIT, and OPP.

In this series of experiments, we opted not to dilute the MWF in TAPS-buffered saline, using sterile filtered tap water instead to better simulate real conditions. Consequently, the pH varied throughout the experiment, dropping by as much as 1 unit in all fluids where *F. solani* grew extensively, indicating a negative influence of fungal presence on MWF chemical stability. In contrast, the experiments involving OPP showed minimal changes in pH, remaining within 0.1 units.

## 4. Discussion

Examining the occurrence data presented in Table 1, we might conclude that fungi are only sporadically present in metalworking fluids (MWFs) and their systems. However, this conclusion is mitigated by two factors. (i) For fungi to be detected, spores or yeast cells must exist in substantial quantities. This is particularly challenging when using cultivation-dependent methods, which have remained standard in the industry. Even if millions of spores are released into the environment, detecting them becomes difficult; spores released into the air are essentially lost, while those in liquid are often heavily diluted and affected by the MWF’s chemistry. (ii) Mold or yeast colonies growing on surfaces frequently go unnoticed, as they can hide within the intricate structures of machines. These organisms only become apparent in obvious locations, such as filters or open tanks, or when their growth disrupts production. Even then, many potential samples are never analyzed because they are simply removed and discarded. Consequently, we think that fungal contamination is widespread in the metalworking industry, without significant concerns.

Publications on fungal contaminations in MWFs are surprisingly scarce and links to disease even more so. This contrasts with the vast literature linking bacteria to occupational diseases [4]. In a review, a connection between microbial colonization of MWFs and the symptoms of illness was presented [35], but molds or yeasts were not mentioned in this context. One possible reason might be a simple one: the species detected (*Fusarium* spp., *Diutina* spp., *Candida* spp. and *Yarrowia* spp.) are reported to be ubiquitous in nature as with many other opportunistic pathogens [36], and constant contact may desensitize most human beings. In any case, it seems difficult to link the experience of symptoms to an exact cause per se [37]. This applies certainly to MWFs as they are made up of a complex mixture of different chemicals, mixed in with water, and used in a plethora of manufacturing processes [38].

The reason that fungi are unwelcome in MWFs is thus mainly of a technical nature: the breakdown of MWF circulation due to blockages of pumps and filters [39]. Alongside this reason are unpleasant odors, visible growth, and fear of the unknown as soft factors. Phenomena such as pH drops and unstable or destroyed emulsions occur and can be simulated in the laboratory but are extremely rare in the real world, as this requires the presence of enormous quantities of fungal material in relation to the tank size. But even if the biological material is in surplus, some MWFs are still able to withstand and remain technically sound. Yeasts, to our knowledge, are of minor concern as phenomena such as those just described could never be attributed to them. In laboratory trials, quantities of up to 10^6^ mL^−1^ had negligible effects on the physico-chemical parameters in MWFs.

The fact that technical problems do not escalate is often attributed to the fungicides contained as in-drum additives. The legal constraints on these chemicals, however, are increasing. Permitted concentrations in the concentrate, as well as in the in-use products, are already restricted, and it is expected that this will continue [22]. Apart from these challenges, simple addition of these ingredients does not always offer an adequate solution; molds or yeasts in splash zones are only marginally impacted by the chemistry as they avoid direct contact. Moreover, the surrounding chemical composition will influence the time required for fungicidal effects to unfold. Another aspect is the availability of the compounds at any given moment. Although concentrates are formulated so that in-use MWFs contain enough fungicides for protection, large quantities of spores may overwhelm the defenses: the substances are used up, are dragged out, and impoverish with time. Additionally, as we have shown, fungi adapt rapidly to the conditions provided by metalworking fluids, enabling them to thrive even in the presence of fungicides. Furthermore, the effectiveness of these fungicides is limited; NaPT showed no measurable impact on spore viability, while BBIT had only a minor effect on the growth of the organism, and neither of their vapors appeared to inhibit fungal development. As our tests were conducted at concentrations that might be common in 5% emulsions (a low-end concentration frequently used in industry), it will be interesting to follow future developments as approved fungicide concentrations continue to decrease. Preliminary trials seem to indicate that halving the concentration already cancels out effectiveness against *Fusarium* spores.

## 5. Conclusions

In the future, the industrial sector needs to adjust by creating MWFs with chemical constructions that deactivate spores and inhibit the growth of molds (and yeasts) without the help of fungicides. Creating an environment that restricts the availability of food sources could be one way to go, as the other important features for fungal growth, i.e., temperature, humidity and pH, cannot be influenced or changed. Indeed, we argue that the ecology created by the chemical composition of the concentrates diluted in water is of higher importance than the anti-fungal activity of single components.

## Figures and Tables

**Figure 1 pathogens-13-00990-f001:**
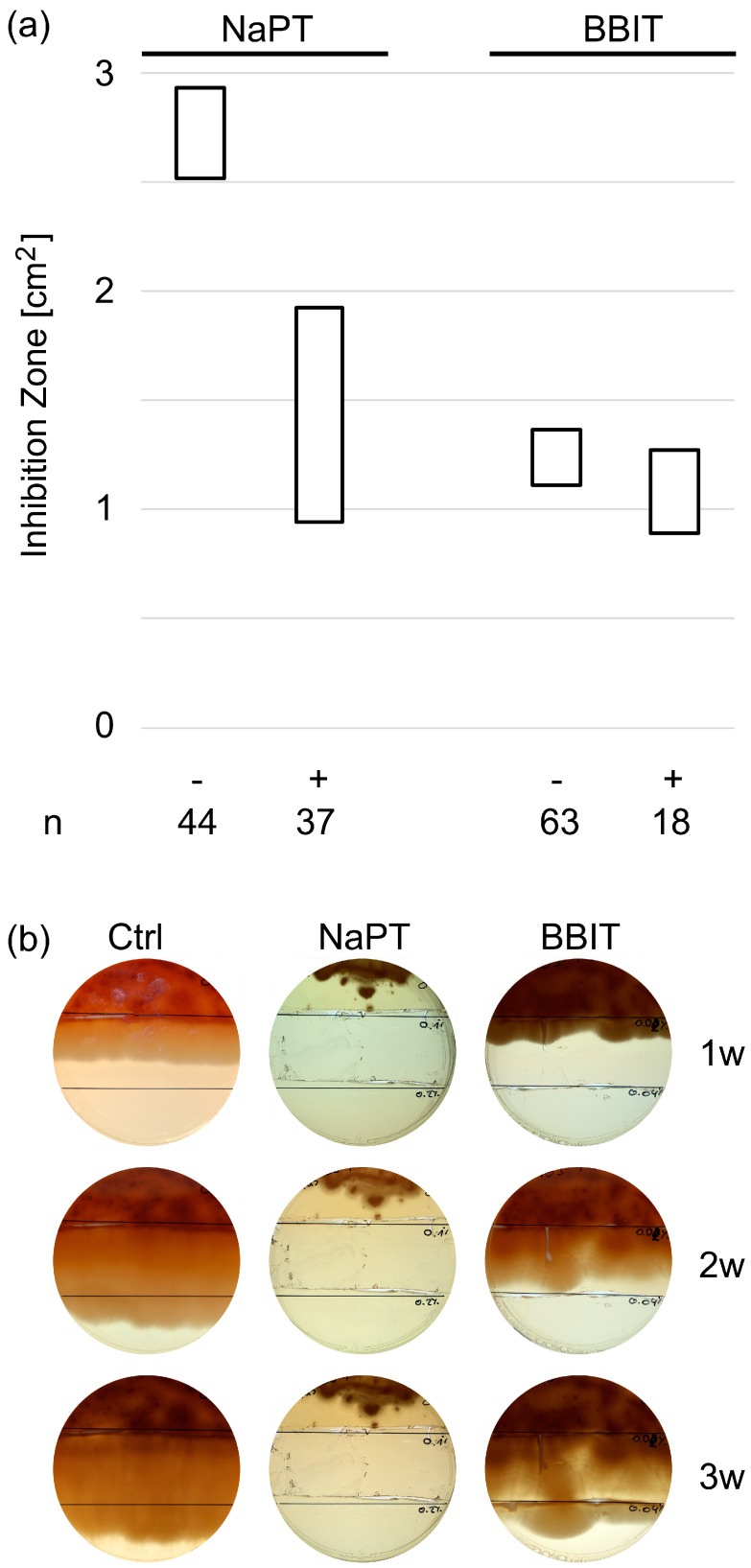
The 95% confidence intervals of the average zone of inhibition formed after 103 fungal samples isolated from in-use MWFs were tested for susceptibility to BBIT or NaPT, whether they had been previously exposed to these substances (+) or not (−) (**a**). Resistance formation was tested in real time using assembled SDA plates containing no fungicides (1. third), a single (2. third) or double dose (3. third) of NaPT (0, 250 ppm, 500 ppm) or BBIT (0, 50 ppm, 100 ppm) (**b**). A typical experiment of three replicates is shown.

**Figure 2 pathogens-13-00990-f002:**
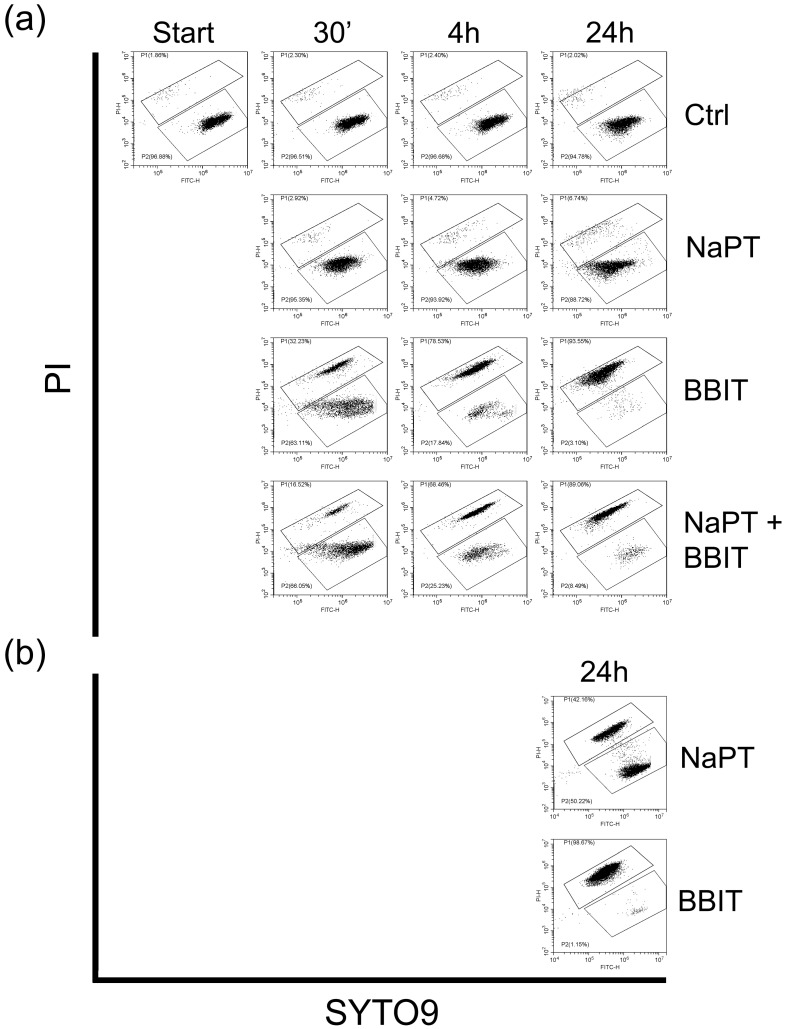
Flow cytometric SYTO9/PI plots presenting the viability of *F. solani* spores at various time points of exposure to TAPS-buffered saline (Ctrl), NaPT, BBIT or both (**a**). The upper region P1 corresponds to the subpopulation of cells with compromised membranes and the lower region P2 to the subpopulation of cells with intact membranes. Tests were also performed with the additional presence of *F. solani* grown on a 1.56 cm^2^ piece of SDA (**b**).

**Figure 3 pathogens-13-00990-f003:**
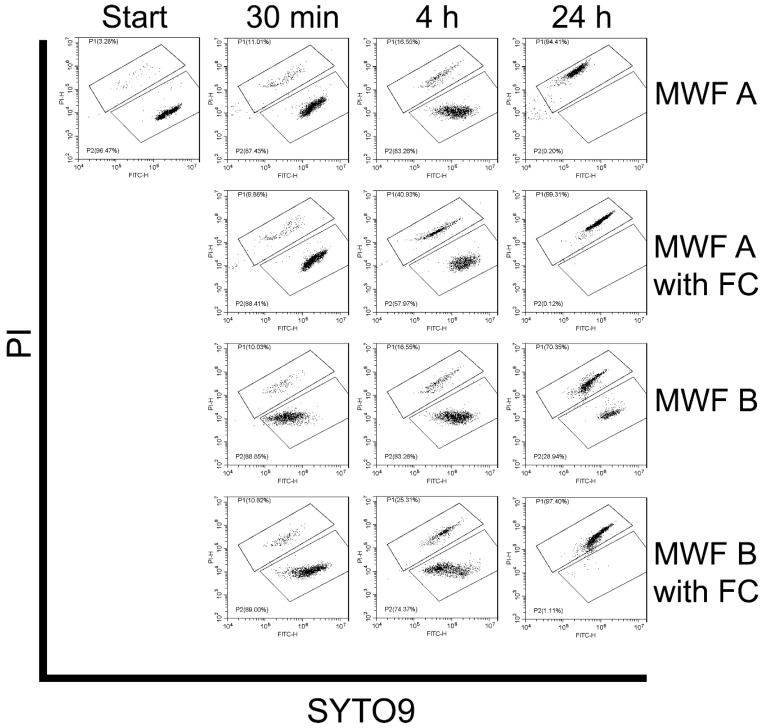
Flow cytometric SYTO9/PI plots presenting the viability of *F. solani* spores at various time points of exposure to experimental MWFs. The upper region P1 corresponds to the subpopulation of cells with compromised membranes and the lower region P2 to the subpopulation of cells with intact membranes. FC: fungicides.

**Figure 4 pathogens-13-00990-f004:**
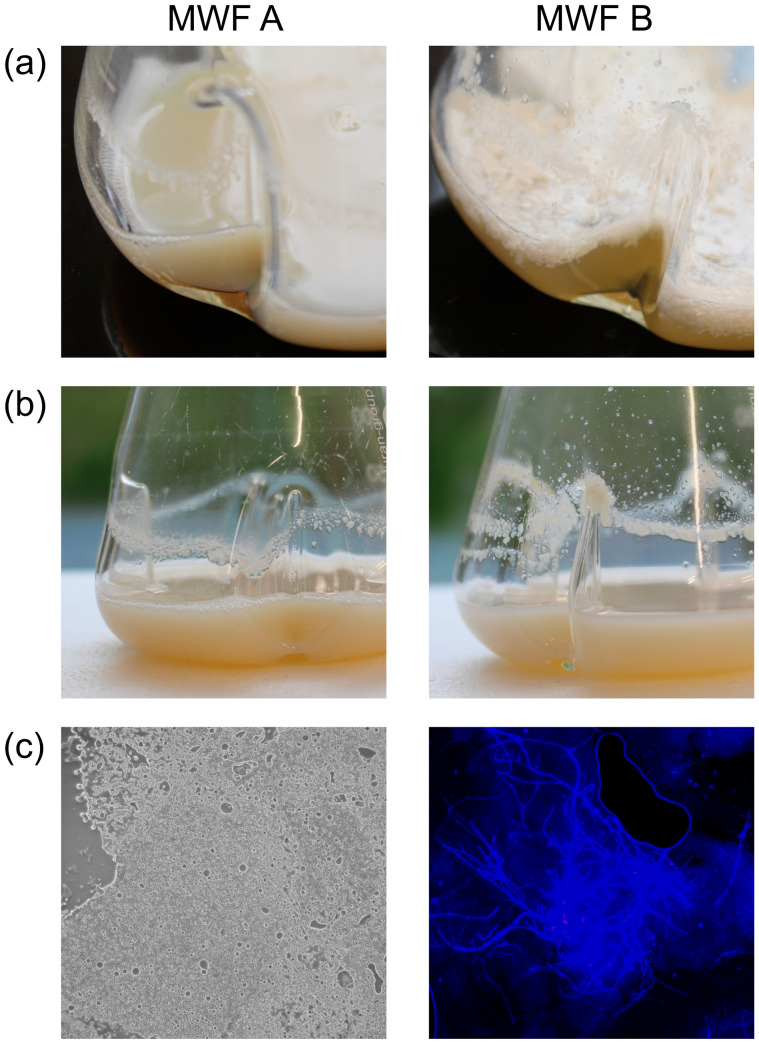
Experimental MWFs exposed to a 1.56 cm^2^ piece SDA overgrown with *F. solani* for three weeks in baffled Erlenmeyer flasks with constant shaking at 80 rpm. Shown are the visual integrity of the emulsion (**a**), the formation of residues at the contact zone (**b**) and the nature of these residues as determined by light and fluorescence microscopy (**c**).

**Figure 5 pathogens-13-00990-f005:**
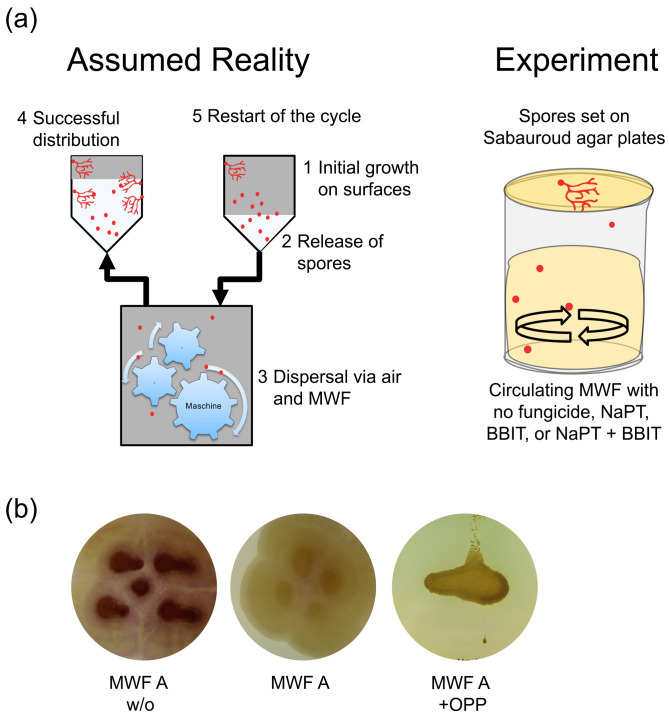
The possible impact of fungicide volatiles was tested in a laboratory trial mimicking reality. The experimental setup attempts to represent the real situation in a highly simplified manner (**a**). Various MWFs with added fungicides were tested as shown, and the effect on the growth of *F. solani* was visually observed over a period of one week (**b**).

**Table 1 pathogens-13-00990-t001:** From 2014 to 2023, a total of 48,695 fluid samples were tested for fungal presence using cultivation-dependent methods, alongside 417 residue samples analyzed by microscopy. Liquid and residue samples were deemed positive when at least 100 cfu mL^−1^ was detectable and deposits in several sub-samples showed fungal cells, respectively.

Year	Fluid Samples		Residue Samples	
	n	Positives	n	Positives
2014	5223	280 (5.4%)	45	17 (37.8%)
2015	5188	387 (7.5%)	27	10 (37.0%)
2016	5053	289 (5.7%)	52	31 (59.6%)
2017	6399	325 (5.1%)	61	44 (72.1%)
2018	5882	348 (5.9%)	40	18 (45.0%)
2019	5131	392 (7.6%)	48	32 (66.7%)
2020	3791	248 (6.5%)	38	22 (57.9%)
2021	3897	156 (4.0%)	33	15 (45.5%)
2022	4229	131 (3.1%)	37	21 (56.8%)
2023	3902	190 (4.9%)	36	19 (52.8%)
Total	48,695	2746 (5.6%)	417	229 (54.9%)

**Table 2 pathogens-13-00990-t002:** Fungi were isolated from 103 in-use samples and subsequently specified by Maldi-TOF MS.

Genus	n	Species	n
*Fusarium*	45	*F. solani*	21
		*F. keratoplasticum*	7
		*F. nierenbergiae*	6
		*F. languescens*	4
		*F. petroliphilium*	4
		*Fusarium* spp.	3
*Paecilomyces*	1	*P. lilacinus*	1
*Diutina*	28	*D. neorugosa*	17
		*D. rugosa*	11
*Candida*	9	*C. tropicalis*	3
		*C. haemulonii*	3
		*Candida* spp.	3
*Yarrowia*	18	*Y. lipolytica*	18
Unknown	2		

**Table 3 pathogens-13-00990-t003:** Released quantities of spores as assessed by flow cytometry in TAPS-buffered saline with added fungicides, MWFs or no addition (Ctrl), as indicated. FC: fungicides.

System	Spore Quantity [Log_10_ mL^−1^]
Buffer (Ctrl)	6.22 ± 0.32
NaPT	6.16 ± 0.42
BBIT	6.19 ± 0.31
NaPT + BBIT	6.21 ± 0.34
MWF A	6.29 ± 0.35
MWF A with FC	6.20 ± 0.32
MWF B	6.13 ± 0.25
MWF B with FC	6.01 ± 0.39

## Data Availability

The original contributions presented in this study are included in the article. Further inquiries can be directed to the corresponding author.

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
