# Peer review of "Fusarium spp. in Metalworking Fluid Systems: Companions Forever"

_pathogens, 2024, doi:10.3390/pathogens13110990_

Round 1
Reviewer 1 Report
Comments and Suggestions for Authors
The paper reports the result of a molecular study on the occurrence of Fusarium app. in metalworking fluid systems. The topic is of interest and the paper is at all well written. Some shortcomings, mostly referred to mycological terminology. Some information about Fusarium solani species complex should be given in introducytion and discussion sections
Line 53 – “molds are important” or “mold state is”
Lines 62-63 – Fungal species belonging to genus Aspergillus., Fusarium., Exophiala., Trichoderma, Cladosporium. and Penicillium have been recovered from MWFs. Please, pay attention to taxonomic groups (i.e. Aspergillus spp. are species, not genera, some of recovered fungi belong to phyla not to taxa)
Line 70 - like Fusarium solani specie complex (FSSC)
Line 71 – please add “as well in animal health, mostly of aquatic species. For these reasons FSSC has been included within pathogens with One Health issue”. Please, cite paper doi 10.3390/jof6040235
Line 72 – fungal contamination (infestion is mostly referred to arthropods)
Line 214 – belonged to genus Fusarium. Please delete spp. Species are contained within genera
Line 219 – (formerly Candida rugosa). Diutina rugosa is a complex of species, also and includes about 0.2% of clinical isolates (DOI: 10.1080/00275514.2019.1585161 ), Candida tropicalis is one of most frequently involved Candida non albicans pathogens
Table 2 and line 259– spp. not italics
Reviewer 2 Report
Comments and Suggestions for Authors
Comments:
The submitted paper ”Fusarium spp. in metalworking fluid systems: Companions forever." by Célia Ruiz and colleagues is interesting and it is very well constructed.
The paper is clear, well written and well organised. Besides, the paper comes out from an industrial company, and not form the Academy, which is very interesting and meritorious in my opinion and not so common. It shows us a different perspective!
The tittle was well chosen! It is a very attention-grabbing tittle indeed!
The abstract is clear, pointing out the main results. Also, keywords were well chosen.
The Introduction is well written, well organised, and the objectives are clearly indicated. It is a complete introduction that represents an excellent state of the art.
Line 63. If the authors refer Aspergillus spp., etc., they do not need to refer the word “genus” (plural genera). If they prefer to use genus or genera, so delete “spp.”.
Methods were appropriate and are well explained with sufficient details. The used techniques were adequate to fully respond to the aims of the study.
But I have a question/doubt: In the results the authors say that: “Nevertheless, the majority of isolated or detected species originated from MWFs containing one or even two fungicides, most often BBIT and/or NaPT.” I mean, I see no detection of fungicides in the methods section…
I realize that you worked with the fungicides commonly detected and used (NaPT and BBIT), standard fungicides used in the industry. But my question is: did you identify these two fungicides in the samples? Or you used those two because of its general use in this area? It is just a detail…
Moreover, I have a question about the fungi identification. Is Maldi-TOF MS analysis of protein patterns sufficient for the identification of Fusarium species, a genus with more than one hundred species? And related to this, it would be important perhaps to do the same studies with a Fusarium solani strain from a reference collection (sourced from a different environment and matrix, for example, plants) for comparison. It is just an idea for the future!
Results
Results are in general well described, and illustrated with the suitable number of figures and tables. Results are clearly presented and are important to the state of the art in metalworking fluids and its biodeterioration. Also, the analyses of the results were properly made.
I just want that the question of the detection of the fungicides to be better explained (Lines 227-230). Please insert a dot after “growth” in line 230.
For me, the results about the resistance and adaptation to fungicides are very important!
Discussion is well articulated and organised.
At the end of the discussion, I do think that the conclusions that are expressed there should be highlighted or even separated in a subsection named “Conclusions”.
The list of references is fine to me!
